# Characterization of Cassava Starch Extruded Sheets Incorporated with Tucumã Oil Microparticles

**Priscila Dayane de Freitas Santos** [1] **, Larissa do Val Siqueira** [2] **, Carmen Cecilia Tadini** [2,3] **and Carmen Sílvia Favaro-Trindade** [1,*]

1 Departamento de Engenharia de Alimentos, Faculdade de Zootecnia e Engenharia de Alimentos, Universidade de São Paulo, Pirassununga 13635-900, Brazil
2 Escola Politécnica, Departamento de Engenharia Química, Universidade de São Paulo, São Paulo 05508-010, Brazil
3 Food Research Center (FoRC/NAPAN), Universidade de São Paulo, São Paulo 05508-080, Brazil
* Correspondence: carmenft@usp.br

**Abstract:** The application of biopolymers and feasible technologies to obtain sheets is crucial for the large-scale production of food packages and for reducing plastic pollution. Additionally, the inclusion of additives in sheets can affect and improve their properties. This work aimed to incorporate tucumã oil (TO) and TO microparticles produced by spray drying (SD), spray chilling (SC), and their combination (SDC) into extruded cassava starch sheets and to evaluate the effect of such addition on their physical, optical, and mechanical properties. Gum Arabic and vegetable fat were used as wall materials for SD and SC/SDC, respectively. The sheets enriched with tucumã oil (FO) and the microparticles produced by SD, SC and SDC (FSD, FSC, and FSDC, respectively) presented yellow color (hue angle around 90°) and higher opacity (11.6–25.3%) when compared to the control (6.3%). All sheets showed high thickness (1.3–1.8 mm), and the additives reduced the water solubility of the materials (from 27.11% in the control to 24.67–25.54% in enriched samples). The presence of large SDC particles, as evidenced by Scanning Electron Microscopy (SEM), caused discontinuity of the sheet structure and decreased mechanical strength of the FSDC. One may conclude that potential active packages were obtained by extrusion of cassava starch sheets added with pure and encapsulated TO.

**Keywords:** extrusion; biopolymer; spray drying; spray chilling; carotenoids; active packaging





## 1. Introduction

Adequate packaging is an important tool for food preservation since its main function is to protect the product from the environment, avoiding contact with factors that can reduce food shelf-life, including moisture, light, and microorganisms, among others [1]. Depending on the thickness of the package layer, packaging materials can be classified as films (thickness < 0.25 mm) or sheets (thickness > 0.25 mm) [2]. So far, synthetic, petroleum-derived plastics have been widely applied for that purpose due to their low-cost and versatile properties, but concerns about their impact on nature have nurtured the search for eco-friendly, biodegradable polymers for packaging applications [3].

Starch is one of the most abundant biopolymers in nature, as the source of energy stored by plants, and it is a suitable option for producing bio-based food packaging due to its high availability, low cost, biodegradability, and film-forming capacity [4]. Among the commercially available types, cassava starch presents low gelatinization temperature and good gel stability and provides non-toxic, colorless, odorless and tasteless films [5]. For this reason, it has been extensively applied in recent years, alone or combined with other polymers, for the formulation of edible [5,6], active [6,7] and smart [8,9] packages. However, as for most biodegradable polymers, the hydrophilic character of starch results in films with poor integrity, and mechanical and barrier properties, features that make their application difficult for packaging purposes [10].

Incorporation of hydrophobic materials, e.g., oils and lipid particles, into starch-based films and sheets is an alternative to overcome these limitations [11–14]. In addition, the enrichment of sheets with bioactive compounds, e.g., carotenoids, could bring supplementary benefits to product attractiveness, such as changes in color and preservation, including antioxidant capacity and antimicrobial activity, among others. Materials that offer additional protection by interacting with the food are known as active packaging [15]. In this sense, tucumã oil (TO) could be an interesting option as an additive for active packages due to its high carotenoid content. The oil, with an intense orange color, is extracted by cold pressing from the pulp of the tucumã fruits (*Astrocaryum vulgare* Mart.), which can be found in the northern region of Brazil, especially in Pará state [16]. Tucumã fruits and TO present a range of health-related effects, such as antioxidant, anti-inflammatory, anticarcinogenic, and antihyperglycemic activities [17,18]. Unfortunately, both the fruit and oil have been poorly explored at the research and industrial level, and the capacity of such materials to contribute to food preservation was not investigated so far. However, β-carotene is known to present antioxidant activity in food systems and has been applied for the production of active films, resulting in antioxidant packages, which improved the oxidative stability of sunflower oil and fried peanuts during storage [6,19,20]. In addition, β-carotene is a vitamin A precursor [21], and its application into edible coatings could improve the nutritional value of the packaged product, as suggested by recent works in which edible films enriched with vitamin $D_3$ were obtained [22,23]. Considering that β-carotene is the major carotenoid in TO [24] and that this compound presents provitamin A activity and strong antioxidant capacity, the addition of TO to films and sheets has the potential to provide similar active and functional properties to food packages.

Production of films by casting technique has been performed to a great extent at the research scale, mainly because it is a simple method that does not require specific and expensive equipment. Nevertheless, the scale-up of this process to the industrial level is limited by its time-consuming, high cost, and low yield character. In this sense, polymeric films and sheets can also be obtained by extrusion, a process widely used in the food and plastic industries [25]. Though incorporation of oil and/or bioactive compounds to film formulations is challenging since both can be oxidized and degraded during extrusion, which is based on the use of high temperature, pressure, and shear to promote polymer melting. Therefore, microencapsulation of sensitive additives is an alternative to enable their application into extruded films and sheets, maintaining their characteristics [26].

In general, encapsulation aims to protect the encapsulated material from degrading conditions by creating a layer around it or by incorporating the compound within a continuous matrix, depending on the applied method [27]. Microencapsulation of carotenoid-rich oils has been carried out through several techniques in recent years [28]. From these techniques, spray drying (SD) stands out as the most common method to produce microparticles, as it is a simple, fast, inexpensive, and continuous process. It is based on the atomization of a water-based dispersion or emulsion inside a chamber with hot air circulation, promoting water evaporation and the formation of dry particles [29,30]. This implies that the obtained particles possess a hydrophilic character. Hydrophobic particles may be produced by a technique similar to spray drying, known as spray chilling (SC). In this process, the carrier material is composed of molten fat, which is mixed with the core and atomized inside a cold chamber. At low temperatures, the molten carrier solidifies around the core, resulting in solid lipid microparticles [31]. The combination of spray drying and spray chilling processes allow the production of particles composed of polymeric cores and lipid wall materials, with unique properties and higher protection of the encapsulated compound [32,33].

The addition of microparticles to film formulations may reinforce the polymer matrix, resulting in materials with improved mechanical properties [34]. So far, no examples of the addition of carotenoid-rich oils, either free or microencapsulated, to biopolymer-based extruded sheets have been found in the literature. In one study, the authors reported the enrichment of cassava starch films, obtained by casting, with carotenoids extracted from

palm oil, but the oil itself was not included in the film matrix [35]. Thus, this work aimed to incorporate TO and TO microparticles, obtained by SD, SC, and the combination of both methods (SDC), into extruded cassava starch sheets and to evaluate the effects of such incorporation on the physical, optical, and mechanical properties of the sheets.

## 2. Materials and Methods

### 2.1. Materials

TO, obtained by a cold-pressing process, was provided by Amazon Oil (Ananindeua, Brasil). Gum Arabic (Nexira, Somerville, NJ, USA) and AL HOME P54 vegetable fat (Cargill, Itumbiara, Brasil), with a melting point of 54 °C, were used as encapsulating agents in SD and SC processes, respectively. For the production of the extruded sheets, native cassava starch was donated by Cargill (Wayzata, MN, USA) and glycerol P.A. (99.5%) was purchased from Synth (Diadema, Brazil).

### 2.2. Microencapsulation of TO

#### 2.2.1. Spray Drying

To obtain TO microparticles by SD, an Arabic gum solution (267 g/kg) was obtained by mixing the powder with distilled water under magnetic stirring. After complete dissolution, the mixture was kept at 7 °C for at least 16 h to ensure gum hydration. Then TO was incorporated into the solution in a proportion of 150 g/kg of total solids. After agitation at 10,000 rpm for 3 min (IKA T25 digital, IKA Brasil, Campinas, Brazil), the resulting emulsion was atomized in a pilot scale spray dryer (MSD 5.0, Labmaq do Brasil Ltd.a, Ribeirão Preto, Brazil), under the following process parameters [17]: 0.5 mm atomizer nozzle, 10 mL/min feed rate and inlet air temperature of 120 °C. Microparticles were stored at −20 °C until further analysis.

#### 2.2.2. Spray Chilling

TO microparticles containing 150 g oil/kg were produced by SC according to the conditions described by Pelissari et al. (2016) [36]. Initially, the vegetable fat was melted at 70 °C and maintained at this temperature for at least 20 min. After that, the core and wall materials were homogenized at 10,000 rpm for 3 min (IKA T25 digital, IKA Brasil, Campinas, Brazil). The mixture was atomized in an adapted spray dryer (MSD 5.0, Labmaq do Brasil Ltd.a, Ribeirão Preto, Brazil) at 14 °C, using a feed rate of 40 mL/min. The microparticles were collected and kept at −20 °C.

#### 2.2.3. Spray Drying Followed by Spray Chilling

The two encapsulation methods described previously were combined to obtain TO microparticles with 22.5 g oil/kg. Microparticles produced by SD were dispersed in the melted vegetable fat (150 g microparticles/kg) and agitated at 6000 rpm for 1 min. The mixture was pumped at 40 mL/min inside a chamber at 14 °C using an adapted spray dryer (MSD 5.0, Labmaq do Brasil Ltd.a, Ribeirão Preto, Brazil). The resulting microparticles were stored at −20 °C.

### 2.3. Total Carotenoid Content and Carotenoid Retention of Microparticles

The total carotenoid content of TO microparticles obtained by the different encapsulation methods was evaluated by adapting the extraction procedures described by Carmona et al. (2018) [37] and Santos et al. (2021) [17].

For SD particles, about 0.045 g of powder were rehydrated with 2 mL of potassium chloride solution (1 g/100 mL) and vortexed (Multi Reax, Heidolph Instruments, Schwabach, Germany) for 3 min. Ethanol (2 mL) was added to the mixture, which was agitated again for 30 s. Carotenoids were then extracted with 3 mL of petroleum ether. For that, the samples were vortexed for 1 min and kept in a sonicating water bath (USC-1400, Unique, Indaiatuba, Brazil) for 5 min. After centrifugation (5430R, Eppendorf, São Paulo, Brazil)

at 4930× *g* for 5 min, the organic upper layer was transferred to another tube, and the extraction process was repeated.

To determine the number of carotenoids in the particles produced by SC, 0.045 g of powder was dissolved in 6 mL of petroleum ether.

Regarding samples obtained by the combination of the two encapsulation techniques (SDC), 0.200 g of particles were initially mixed with 3 mL of petroleum ether and vortexed for 5 min. At the end of this time, 2 mL of potassium chloride solution (1 g/100 mL) and 2 mL of ethanol were added to the tubes, which were stirred for three more min. The mixtures were sonicated and centrifuged at 4930× *g* for 5 min. After the collection of the supernatant, the extraction procedure was performed one more time.

The absorbance of petroleum ether extracts was measured using a UV-Vis spectrophotometer (Genesys 10s, Thermo Scientific, Waltham, MA, USA) at 450 nm, and the total carotenoid content, in mg/kg, was calculated through Equation (1) [38]:

$$\text{Total carotenoid content (mg/kg)} = (A \times V \times 10000)/(2592 \times m) \tag{1}$$

wherein A = absorbance, V = petroleum ether volume (mL), 2592 = β-carotene absorption coefficient in petroleum ether and m = mass of sample (g). The analyses were carried out in quadruplicate.

Carotenoid retention after the encapsulation processes was expressed as a percentage and calculated by comparing the total carotenoid content of TO microparticles and the theoretical carotenoid content according to each formulation (based on the carotenoid content of pure TO and the oil loading in the particles).

### 2.4. Obtaining Cassava Starch-Based Sheets by Extrusion

The production of extruded sheets incorporated with encapsulated and non-encapsulated TO occurred in two steps, following the method reported by Vedove et al. (2021) [8]. The first one was the extrusion of pellets, for which pre-mixtures composed of cassava starch, glycerol, and water (52.5:25.0:20.5, dry basis), were added with non-encapsulated TO and with the microparticles SD, SC and SDC in the proportions 0.15, 1.00, 1.00 and 7.00% (*w/w*, dry basis), respectively. These proportions were selected to reach similar total carotenoid contents in the final products (considering the carotenoid content in non-encapsulated TO and the different oil loadings of particles SD, SC and SDC). The treatments, including a control, were mixed in a domestic blender (Stand Mixer 525, KitchenAid, Benton Harbor, MI, USA) at speed 2 for 30 min. At the end of this time, the mixtures were fed into a co-rotating twin-screw extruder (L/D 40, bolt diameter of 11 mm and die diameter of 3 mm; Process 11, Thermo-Fisher Scientific, Karlsruhe City, Germany) at 0.06 kg/h, using a twin-screw feeder (Volumetric Minitwin Process 11, Brabender, Karlsruhe, Germany). The screw speed was 80 rpm, and the barrel was composed of eight heating zones (60, 75, 90, 100, 110, 110, 105 and 100 °C). The extruded material was kept at room temperature for 24 h and then manually cut into small and uniform pellets.

The second step was the extrusion of sheets, which was carried out under the same process conditions described above. The pellets were fed into the twin-screw extruder at approximately 0.1 kg/h and subjected to a ninth heating zone at 105 °C. The dimensions of the die were 27.0 mm width × 1.0 mm height. After extrusion, the sheets were kept at room temperature for 24 h and then packed in sealed plastic bags under vacuum until further analysis.

### 2.5. Characterization of the Sheets

#### 2.5.1. Opacity and Color Parameters

The opacity, lightness (*L\**), redness (*a\**), and yellowness (*b\**) of the control sheet and sheets incorporated with encapsulated and non-encapsulated TO were evaluated using a colorimeter (MiniScan XE Plus, HunterLab, Reston, USA), according to the CIELAB color

system. Before analysis, the equipment was calibrated with black and white standards. From these parameters, the hue angle ($H°$) and chroma ($C*$) were calculated as follows:

$$C^* = \sqrt{(a^*)^2 + (b^*)^2} \tag{2}$$

For $H°$, the right equation to be used depends on which quadrant of the CIELAB space the $a*$ and $b*$ coordinates are located. So, for [$+a*$, $+b*$], Equation (3) was applied, and for [$-a*$, $+b*$], Equation (4) was used [39]:

$$H° = \tan^{-1}\left(\frac{b^*}{a^*}\right) \tag{3}$$

$$H° = 180 + \tan^{-1}\left(\frac{b^*}{a^*}\right) \tag{4}$$

The total color difference ($\Delta E$) of sheets after storage at 25 °C for 60 days was obtained through Equation (5), according to Liu et al. (2016) [40]. This parameter was used as an indirect indicator of carotenoid degradation in the sheets.

$$\Delta E = \sqrt{(\Delta L^*)^2 + (\Delta a^*)^2 + (\Delta b^*)^2} \tag{5}$$

### 2.5.2. Thickness and Mechanical Properties

The thickness of cassava starch-based sheets, enriched or not with encapsulated and non-encapsulated TO, was measured at different positions using a micrometer (ID-C112PM, Mitutoyo, Kawasaki, Japan) and presented as mean ± standard deviation of ten measurements.

The tensile strength and elongation at the break of the sheets were evaluated in a texture analyzer (Texture TAXT2i, Stable Micro Systems Ltd., Surrey, United Kingdom) according to the conditions reported by Vedove et al. (2021) [8]. Initially, all samples were maintained at 23 °C and 50% relative humidity for 14 days. At least six sheet strips of each treatment, measuring 100 mm in length and 10 mm in width, were subjected to extension, with an initial grip of 50 mm and a crosshead speed of 0.42 mm.s$^{-1}$. Only samples that broke far from the grips were considered for calculations. Young's modulus was determined as the slope of the initial linear portion of the obtained curves.

### 2.5.3. Moisture Content

The moisture content of extruded sheets was evaluated according to Mei et al. (2013) [41]. For that, samples (approximately 3 g) were maintained at 105 °C for 24 h and weighed again at the end of this time. The moisture content (%) was determined as follows:

$$MC = \frac{w_i - w_f}{w_i} \times 100 \tag{6}$$

wherein $w_i$ is the initial weight of the sample and $w_f$ is its final weight, in g. The analyses were carried out in triplicate.

### 2.5.4. Microstructure of Films

Scanning electron microscopy (TM3000 Tabletop Microscope, Hitachi, Tokyo, Japan) was performed to evaluate the cross-section morphology of the samples under an accelerating voltage of 5 kV. Before imaging, sheets were frozen in liquid nitrogen and then fractured with a sharp blade.

### 2.5.5. Water Solubility

The method described by Samsalee & Sothornvit (2020) [42] was carried out to assess the solubility of sheets in water. For that, sample squares with 400 mm$^2$ of area (20 mm length × 20 mm width) were immersed in 40 mL of water and kept at 25 °C for

24 h, under agitation. At the end of this time, the sheet pieces were removed from the water and dried at 105 °C for 24 h to obtain their final dry weight. The water solubility of sheets (%) was calculated by comparing their initial and final dry weights. Analyses were performed in triplicate.

*2.6. Statistical Analyses*

Data were analyzed using Statistica software (STATISTICA, version 13.5.0.17, TIBCO Software Inc., Palo Alto, CA, USA.). ANOVA and post hoc Tukey's tests were performed, with a significance level of 0.05.

## 3. Results and Discussion

*3.1. Characterization of TO Microparticles*

Pure TO and TO microparticles produced by SD, SC and SDC were characterized regarding their total carotenoid content (TCC) and carotenoid retention after encapsulation, as can be seen in Table 1. Non-encapsulated oil presented a high amount of carotenoids, much greater than the values reported in the literature for the same material [43,44]. In fact, Santos et al. (2021) [17] recently quantified total carotenoids of TO spectrophotometrically and found the content of 511 $\mu g/g_{oil}$, revealing that the composition of tucumã fruits, and consequently of the extracted oil, is largely affected by the environment and growing conditions, even when the oil is provided by the same company. Regarding the encapsulated oil, microparticles showed much lower TCC, as expected, due to a dilution effect since only 15% of the oil was used as the core in the particles prepared by SD and SC, and an even smaller percentage of oil was incorporated into the final product after a combination of the techniques. Notwithstanding, the carotenoid content of all microparticles can still be considered high in comparison to similar materials. For example, red pepper waste extract encapsulated by spray drying presented a carotenoid content of ~32 $\mu g/g$ [45]. Neither SC nor SDC has been used for the encapsulation of carotenoid-rich materials so far.

**Table 1.** Total carotenoid content of tucumã oil and microparticles obtained by different techniques, as well as carotenoid retention after encapsulation.

| Sample | Total Carotenoids (μg/g) | Carotenoid Retention (%) |
|:------:|:------------------------:|:------------------------:|
| TO | 2679.1 ± 217.1 [a] | - |
| SD | 366.0 ± 4.0 [b] | 91 ± 1 [a] |
| SC | 338.9 ± 9.9 [b] | 84 ± 2 [b] |
| SDC | 55.0 ± 0.7 [c] | 91 ± 1 [a] |

TO: tucumã oil; SD: spray drying; SC: spray chilling; SDC: spray drying combined with spraying chilling. Values are mean ± standard deviation (*n* = 4). Different letters in the same column indicate significant differences (*p* < 0.05).

To understand the effect of each encapsulation process on pigment degradation, carotenoids were also expressed in terms of retention in the particles. All samples exhibit high retention of these bioactive compounds, meaning that small carotenoid losses occurred during microparticle production. Interestingly, SC caused more pigment degradation than SD, which is surprising since the first is carried out at a low temperature, while the latter requires heating above 100 °C. Considering that carotenoids are sensitive to oxidation and that high temperatures trigger and accelerate such reactions [38], retention was expected to be lower in SD samples. This may be due to the heating step necessary to melt the wall material (vegetable fat) and keep the infeed dispersion liquid for SC since this dispersion is maintained at approximately 70 °C for a few minutes before atomization. However, the carotenoid retentions found here are higher than in most recent works on microencapsulation of carotenoid-rich materials by varying techniques, as reviewed in a previous work of our research group [28].

The improved stability of sensitive bioactive-like carotenoids after encapsulation is crucial for the later application of the obtained particles. As mentioned above, carotenoids are mainly degraded by oxidation, which is triggered by high temperatures, oxygen, and

light, among other factors [38]. In addition, β-carotene is the major bioactive compound in tucumã oil that could provide active properties to sheets for food packaging purposes, as discussed before. Previous studies demonstrated the impact of encapsulation on preserving the properties of materials with relevance in terms of food preservation. For example, spray drying improved the oxidative stability of tucumã oil during exposition to high temperatures (similar to the ones applied during extrusion) by at least 142-fold, suggesting that encapsulation maintained the oil/carotenoids properties of interest for food preservation [17]. In another work, cassava starch films containing nano encapsulated β-carotene were more efficient in preventing sunflower oil oxidation during storage (peroxide value of 6.72 mEq/kg) compared to non-encapsulated β-carotene (peroxide value of 9.31 mEq/kg), due to the improved stability of the encapsulated carotenoid and the higher light-barrier property of films added with the particles [6].

### 3.2. Characterization of Extruded Cassava Starch-Based Sheet

The incorporation of non-encapsulated oil and TO microparticles into extruded cassava starch sheets resulted in samples with different appearances (Figure 1). The control sheet (FB) looks clear and translucent, as expected, since cassava starch does not contain any pigments, and the color becomes slightly yellow with the addition of pure oil (FO) and SD particles (FSD) to the formulation. All samples presented a rough surface, including those enriched with particles produced by SC (FSC) and the combination of the methods (FSDC), and such characteristic has been observed before for sheets obtained by extrusion [8]. However, FSC and FSDC were more opaque and showed a more intense yellow color than the previous samples, which may be caused by the vegetable fat used as wall material for SC or by higher retention of encapsulated carotenoids after the extrusion processes.

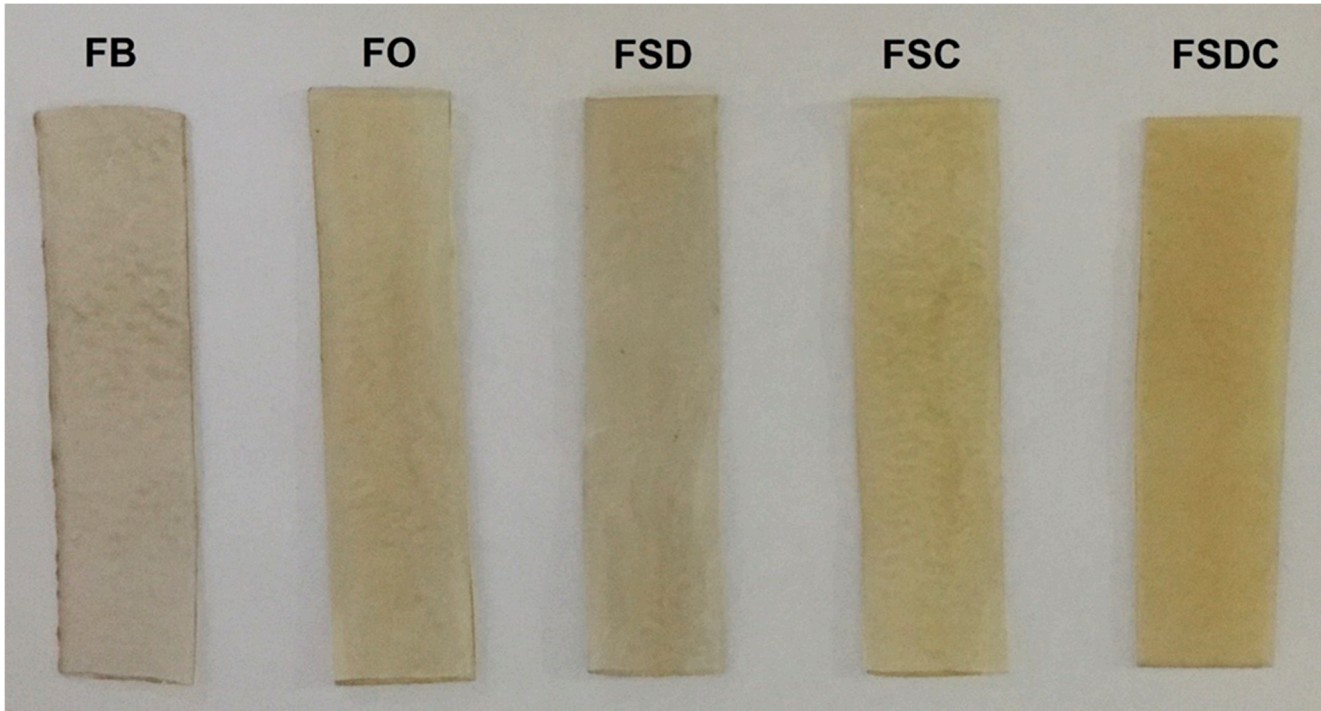

**Figure 1.** Visual aspect of cassava starch sheets enriched or not (FB) with tucumã oil (FO) and tucumã oil microparticles (FSD, FSC and FSDC) obtained by different encapsulation methods.

The visual differences in opacity and color were confirmed by the optical parameters of the sheets, which are presented in Table 2. The addition of non-encapsulated oil or microparticles to cassava starch formulations significantly increased the opacity of sheets. Specifically, FSDC samples were the most opaque among enriched materials. This barrier property is interesting for the protection of light-sensitive foods against degradation [46].

Ghiasi et al. (2020) [14] also observed that including pure sunflower oil or structured oil nanoparticles in Farsi gum-based films increased their opacity by ~4–9 fold, and the authors attributed these changes to light-scattering by additives inside the film matrix, resulting in lower transparency. This could also explain the results found in the present study. Regarding color parameters, the sheets incorporated with non-encapsulated or encapsulated oil all presented high $H°$ values (around 90°) when compared to the control. According to McLellan et al. (1995) [39], angles in this range indicate a yellow color. Such color of the enriched sheets is attributed to the carotenoids found in TO and is an indirect measure of pigment retention after the extrusion process.

**Table 2.** Color parameters of the control sheet (FB) and sheets added with tucumã oil (FO) and microparticles produced by varying techniques (FSD, FSC and FSDC), as well as the total color difference of sheets after storage a room temperature for 60 days.

| Sheet | Opacity (%) | $H°$ | $C*$ | $\Delta E$ |
|---|---|---|---|---|
| FB | 6.3 ± 0.7 [c] | 51.45 ± 7.12 [b] | 1.7 ± 0.2 [d] | - |
| FO | 11.6 ± 1.0 [b] | 91.79 ± 0.18 [a] | 16.1 ± 0.3 [c] | 10.3 ± 2.9 [a] |
| FSD | 11.7 ± 0.8 [b] | 90.78 ± 0.31 [a] | 21.9 ± 1.0 [b] | 11.4 ± 1.0 [a] |
| FSC | 13.8 ± 1.0 [b] | 89.35 ± 0.51 [a] | 24.8 ± 1.4 [a] | 12.6 ± 0.5 [a] |
| FSDC | 25.3 ± 2.6 [a] | 86.33 ± 0.08 [a] | 23.2 ± 0.3 [ab] | 3.8 ± 1.5 [b] |

$H°$: Hue angle; $C*$: Chroma value; $\Delta E$: total color difference. Values are mean ± standard deviation (SD) ($n$ = 3). Different letters in the same column indicate significant differences ($p < 0.05$).

Although the color tone of the enriched sheets was the same, the color intensity/saturation, i.e., chroma, varied significantly with the type of additive. In all cases, the saturations were relatively low, considering, for example, that polylactide (PLA) films containing 0.1% bixin presented an intense yellow color with a chroma value of 105.5 [47]. FO treatment had the lowest saturation, suggesting a light-yellow aspect, while the FSC showed the most intense yellow color among sheets added with TO microparticles, which also corroborates with their visual appearance, as can be seen in Figure 1. The significantly higher chroma of sheets added with microparticles (FSD, FSC and FSDC) compared to that enriched with pure oil (FO) may suggest that encapsulation protected carotenoids from degradation during extrusion at high temperatures. A similar protective effect was already reported in the literature for other bioactive compounds. For example, Chen et al. (2019) [45,48] demonstrated that encapsulation of citral and trans-cinnamaldehyde by the inclusion of the complex in β-cyclodextrin reduced bioactive loss during extrusion, as indirectly evidenced by the higher antimicrobial activity of films enriched with encapsulated compounds.

Nonetheless, microparticles showed varying efficiency in preventing carotenoid degradation during prolonged storage of enriched sheets. As expressed by total color difference ($\Delta E$), changes in the color of samples were significant for all treatments after 60 days at room temperature, which is an indicator of pigment loss over time. Specifically, the $\Delta E$ of the sheets FSD and FSC did not differ from the value found for FO, suggesting that SD and SC particles were probably damaged during extrusion and, thus, were not able to provide long-term stability to carotenoids. This can be confirmed by the great decrease in parameter $b*$ (yellowness) presented by these treatments at the end of storage (data not shown). On the other hand, the color change of the FSDC sheet was much smaller compared to the previous ones, revealing that the combination of SD and SC was efficient in slowing down carotenoid depletion, even after exposure to harsh extrusion conditions. This correlation between color change and carotenoid degradation was evidenced by Tupuna-Yerovi et al. (2020) [49], who studied the color stability of isotonic tangerine soft drinks added with microencapsulated and non-encapsulated norbixin during accelerated storage conditions. They observed that samples showing higher $\Delta E$ values also presented lower norbixin retentions over time. It is possible that the variation in color of the enriched sheets reported here would not be easily perceived by the human eye. In fact, according to Maia et al. (2019) [50], color changes become visually evident when $\Delta E$ is higher than 5. Recently, Stoll et al. (2021) [44,47] reported bigger color variations (21.6–43.8) as a result

of carotenoid degradation after subjecting polylactide films with 0.01–0.1% bixin to heat treatment at 160 °C for 6 min.

In terms of physical characterization (Table 3), cassava starch sheets presented moisture content (MC) of around 19%, which is lower than the values found in the literature for similar cassava starch samples (~22%) and other polysaccharide-based materials, such as chitosan (~25%) and water chestnut starch films (~21%) [41,51,52]. Some studies reported a decrease in the MC of biopolymer-based sheets after the addition of lipids to the formulations [50,51,53,54]. This trend was not observed in the present study since the presence of TO, and particles with vegetable fat as wall material in treatments FO, FSC, and FSDC did not result in significantly different MC values in comparison to the control. Conversely, the inclusion of SD particles in the sheets caused a significant increase in MC, probably because of the hygroscopic nature of gum Arabic, used as the carrier agent to encapsulate TO by this method.

**Table 3.** Characterization of control sheet (FB) and sheets incorporated with tucumã oil (FO) and its microparticles (FSD, FSC and FSDC), obtained by different encapsulation methods, regarding their moisture content (*MC*) and water solubility (*WS*).

| Sheet | *MC* (%) | *WS* (%) |
|:---:|:---:|:---:|
| FB | 18.0 ± 0.3 [b] | 27.11 ± 0.30 [a] |
| FO | 19.5 ± 0.4 [ab] | 24.87 ± 0.11 [c] |
| FSD | 21.2 ± 0.7 [a] | 24.67 ± 0.28 [c] |
| FSC | 19.3 ± 0.6 [ab] | 25.54 ± 0.34 [b] |
| FSDC | 19.4 ± 1.6 [ab] | 24.71 ± 0.05 [c] |

Values are mean ± standard deviation (SD) (*n* = 3). Different letters in the same column indicate significant differences (*p* < 0.05).

Table 3 also shows the results for the water solubility (*WS*) of the extruded sheets. This is an important property of biodegradable food packaging since it directly affects the preservation of food products and the impact of the material on the environment. Herein, the incorporation of non-encapsulated and encapsulated TO reduced the *WS* of cassava starch sheets. According to Wang et al. (2021) [55] who investigated the properties of corn starch films incorporated with *Zanthoxylum bungeanum* essential oil, this reduction of *WS* is expected due to the hydrophobic character of the additives. Moreover, the presence of lipid compounds may have prevented the absorption of water and swelling of starch granules, thus hindering sheet dissolution, as suggested before by other authors [8,56].

Cassava starch sheets presented heterogeneous and high thickness (>1.3 mm), as displayed in Table 4, which is not ideal for application as food packaging. Additionally, the values were higher than usually found in the literature for extruded films. For example, the cassava starch-based sheet produced by Vedove et al. (2021) [8] by extrusion had an average thickness of 0.6 mm. However, the enrichment of the samples with anthocyanin extract increased their thickness to 1.7 mm. In another work, extruded thermoplastic starch films showed 0.9 mm of thickness, and the addition of cellulose acetate and/or a catalyst also provided thicker samples (1.0–1.2 mm) [57]. Herein, while the addition of SC and SDC particles did not significantly affect the thickness of sheets, incorporating pure oil and SD particles resulted in thicker materials. The explanation for these results is not clear, but other authors have stated that the inclusion of additives could create more complex starch matrices and/or increase the solid content of the material and, thus, promote a thickening effect [3,9]. Despite the effect of the additives on this property, even the control cassava starch sheet had an unusually high thickness, revealing that the extrusion process should be further optimized to provide samples with adequate characteristics for packaging purposes. In addition, the thickness of sheets would probably play an important role in their barrier properties, such as oxygen and water vapor permeabilities [2,3,55,56,58,59]. Unfortunately, it was not possible to evaluate these parameters in the obtained samples due to a physical limitation of the extrusion equipment: the sheet-forming mold attached to the extruder die

had a fixed width of 2.7 cm, but wider samples are required for the determination of barrier properties by most methodologies found in the literature.

**Table 4.** Thickness and mechanical properties of a control (FB) and enriched sheets, incorporated with tucumã oil (FO) and tucumã oil encapsulated by spray drying (FSD), spray chilling (FSC) and the combination of both techniques (FSDC).

| Sheet | Thickness (mm) | *TS* (MPa) | *E* (%) | *YM* (MPa) |
|---|---|---|---|---|
| FB | 1.4 ± 0.1 [b] | 2.4 ± 0.1 [a] | 74 ± 4 [a] | 0.260 ± 0.019 [a] |
| FO | 1.7 ± 0.5 [a] | 2.4 ± 0.2 [a] | 57 ± 6 [b] | 0.236 ± 0.033 [a] |
| FSD | 1.8 ± 0.4 [a] | 2.1 ± 0.1 [ab] | 65 ± 7 [ab] | 0.247 ± 0.025 [a] |
| FSC | 1.3 ± 0.2 [b] | 2.3 ± 0.1 [a] | 59 ± 7 [ab] | 0.231 ± 0.016 [a] |
| FSDC | 1.3 ± 0.1 [b] | 1.8 ± 0.1 [b] | 31 ± 4 [c] | 0.259 ± 0.008 [a] |

*TS*: tensile strength; *E*: elongation at break; *YM*: Young's modulus. Values are mean ± standard deviation (SD) ($n = 3$). Different letters in the same column indicate significant differences ($p < 0.05$).

In the present study, no correlation was found between the thickness and mechanical properties of cassava starch sheets enriched with encapsulated and non-encapsulated TO. In fact, only the incorporation of SDC microparticles caused a remarkable decrease in the resistance and flexibility of the samples, as can be seen in Table 4 in terms of the tensile strength (*TS*) and elongation at break (*E*) of the sheets. This could be related to the microscopic aspect of the materials. From Figure 2, it can be noticed that the cross-section of the FSDC sample contains large SDC particles, and fissures originating from one of them can be observed. The presence of these large particles led to discontinuity of the polymer matrix, which probably favored the disruption of the sheet when subjected to mechanical stress. Luchese et al. (2018) [60] also reported a reduction in *TS* and *E* after the inclusion of blueberry residue powder in cassava starch films obtained by casting. According to the authors, the addition of large and heterogeneous particles to the film matrix can lower its cohesion and compactness, resulting in weak points that affect the mechanical properties of the material.

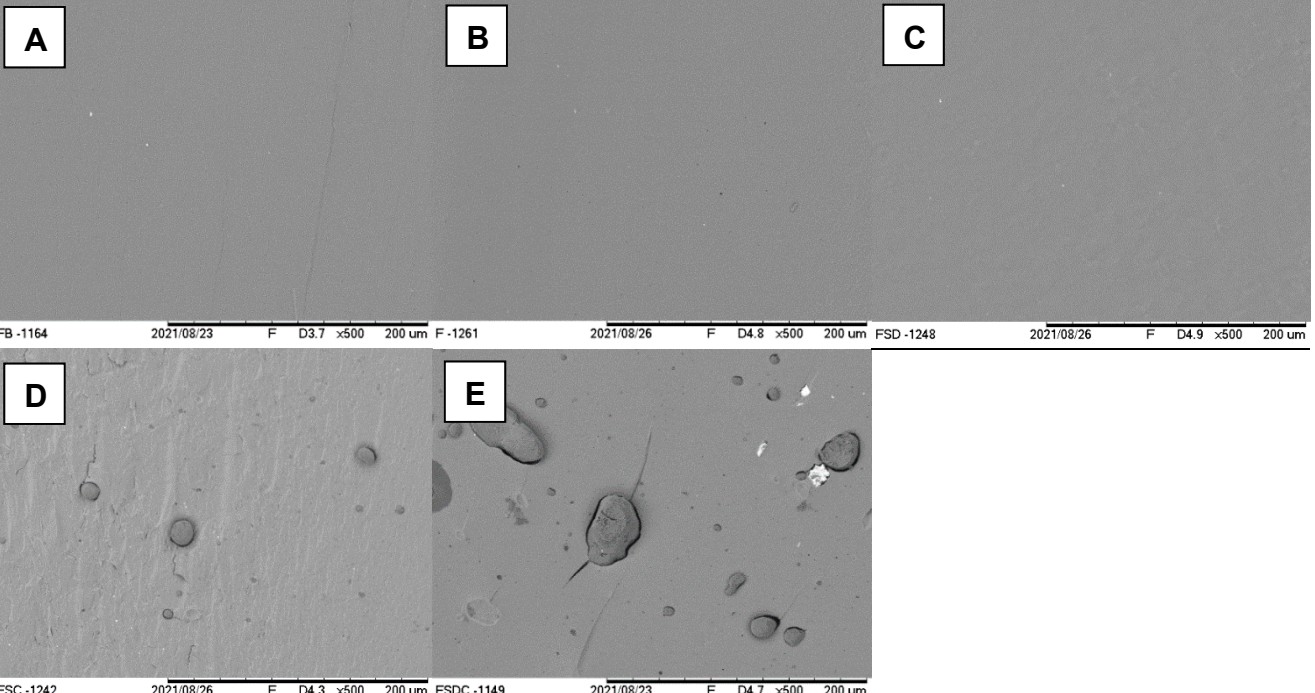

**Figure 2.** SEM micrographs of the control sheet (**A**), sheets incorporated with non-encapsulated (**B**) and encapsulated tucumã oil (**C**–**E**; with particles SD, SC and SDC, respectively), at 200× magnification.

Young's modulus (YM) is a measure of the stiffness, or rigidity, of polymeric materials, such as films and sheets [61]. As can be seen in Table 4, the YM of extruded sheets containing TO and TO microparticles did not differ significantly from the control sample. In addition, the YM values observed here are much lower than those reported in recent works for similar materials. For example, Estevez-Areco et al. (2019) [51] applied extrusion followed by thermo-compression to produce cassava starch films added with rosemary extract, which presented a YM ranging from 9.3 to 24.1 MPa. In another study, cassava starch-PBAT (poly (butylene adipate co-terephthalate)) blown films, with or without *Baccharis dracunculifolia* leaf powder, showed a YM of 28.55–44.22 MPa [62]. Interestingly, there seems to be a correlation between thicker sheets (thickness close to or higher than 1 mm) and lower YM values, considering the results found by Vedove et al. (2021) [8] and Herniou-Julien et al. (2019) [57], suggesting the low modulus observed here could be related to the high thickness of cassava starch sheets.

The mechanical properties of enriched samples indicate that neither pure oil nor microencapsulated oil was able to reinforce the sheets' structure, as opposed to recent studies that reported film reinforcement after the inclusion of microparticles as additives. In fact, the addition of 1 and 5% of β-chitin microparticles to polyvinyl alcohol/chitosan films (70:30, $w/w$) increased *TS* and *E* up to approximately 2.7 and 2-fold, respectively [34]. In this sense, the strengthening or weakening of polymeric films seems to be related to good or poor interaction between materials composing the film matrix and the added microparticles [26], suggesting low compatibility between cassava starch and TO microparticles. As mentioned above, optimization of the extrusion process should be carried out to produce samples with adequate characteristics for packaging applications, and such changes in process parameters could also modify the strength of cassava starch sheets and the way TO microparticles affect their mechanical properties.

Regarding the microstructure of the samples (Figure 2), the control sheet (FC) shows a smooth and homogeneous cross-section, which indicates that starch granules were completely gelatinized during extrusion. The incorporation of TO did not result in a rough and irregular cross-section, as observed by Ghiasi et al. (2020) [14], who added sunflower oil to Farsi gum films and verified oil accumulation on the film surface, producing discontinuous samples. This reveals good compatibility between TO and cassava starch matrix. SD particles also did not affect the microscopic appearance of the sample, suggesting that particles were probably destroyed during sheet production and homogeneously integrated into the polymer network. On the other hand, intact SC and SDC particles are visible in the cross-section of samples FSC and FSDC. This contradicts the hypothesis that high color changes in FSC sheets after storage were caused by the collapse of SC particles during extrusion. The integrity of SDC particles could be related to higher particle resistance to harsh processing conditions, which correlates well with the superior carotenoid stability in the FSDC sheets.

## 4. Conclusions

In this work, cassava starch sheets incorporated with non-encapsulated and encapsulated TO were produced by extrusion, resulting in materials with yellow color and improved light-barrier properties. Furthermore, the enriched sheets showed higher resistance to water, which is a critical packaging feature to ensure food preservation. Though, the extrusion process must be optimized to provide materials with adequate thickness and mechanical properties for practical applications. Moreover, based on the results presented here and on the fact that adding microencapsulated additives to cassava starch sheets elevates their cost (especially considering the microparticles produced by a combination of two encapsulation processes), it would be worth investigating other potential characteristics of the sheets enriched with encapsulated and non-encapsulated TO in future works, such as antioxidant activity, carotenoid migration to food simulants, and application of the sheets as food packages to assess their ability to protect sensitive products from oxidation. Nevertheless, the addition of an oil rich in provitamin A β-carotene, in free and

microencapsulated form, to extruded sheets based on biopolymers is reported here for the first time, and the obtained materials have promising attributes for use as active packaging.

**Author Contributions:** P.D.d.F.S.: Conceptualization, Methodology, Validation, Formal analysis, Investigation, Writing—Original Draft, Writing—Review & Editing, Visualization, Project administration. L.d.V.S.: Conceptualization, Methodology, Investigation, Writing—Review & Editing. C.C.T.: Conceptualization, Methodology, Resources, Writing—Review & Editing, Supervision, Funding acquisition. C.S.F.-T.: Conceptualization, Methodology, Resources, Writing—Review & Editing, Supervision, Funding acquisition. All authors have read and agreed to the published version of the manuscript.

**Funding:** This research was funded by Coordenação de Aperfeiçoamento de Pessoal de Nível Superior (CAPES) (scholarship granted to P.D.d.F.S, no. 88887.571710/2020-00); by the Fundação de Amparo à Pesquisa do Estado de São Paulo (FAPESP) (grant to L.d.V.S., 2019/21700-7) and by the project EMU 2016/12385-2; and by the Conselho Nacional de Desenvolvimento Científico e Tecnológico (CNPq, Brazil) (grants to C.C.Tadini and C.S.F.T., numbers 309548/2021 and 310686/2022, respectively).

**Data Availability Statement:** Data are available on request from the authors.

**Acknowledgments:** The authors would like to thank Coordenação de Aperfeiçoamento de Pessoal de Nível Superior (CAPES) for the scholarship granted to P.D.d.F.S. (No. 88887.571710/2020-00). The authors acknowledge the São Paulo Research Foundation (FAPESP) for the grant to L.d.V.S. (2019/21700-7), the project EMU 2016/12385-2; and the National Council for Scientific and Technological Development (CNPq, Brazil) for the grants to C.C.T. (309548/2021) and C.S.F.T. (310686/2022).

**Conflicts of Interest:** The authors declare no conflict of interest.

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
