# Peer review of "Characterization of Cassava Starch Extruded Sheets Incorporated with Tucumã Oil Microparticles"

_processes, doi:10.3390/pr11030876_

Round 1

Reviewer 1 Report

The manuscript entitled "Characterization of cassava starch extruded sheets incorporated with tucumã oil microparticles" can be improved by addressing the suggestions given below :-

- Line no.- 24, "The sheets enriched with TO (FO) and the micro- 24 particles (FSD, FSC, and FSDC)" ....  the abbreviations are not clear

- Line no.- "One may conclude that promising active packages were obtained by extrusion of cassava 30 starch sheets added with pure and encapsulated TO" ... how authors can say the developed sheet is active??? ...Which active properties they have tested?? ...

- The release of oil from the sheet/oil microparticles should be included and/or any active function of the film

- Figure 2, should be improved by placing 2 or 3 pictures in a row

Line no.- 488 "Nevertheless, the addition of an oil rich in provitamin A β-carotene, in free and microencapsulated form, to extruded sheets based on biopolymers is reported here for the first time, and the obtained materials have promising attributes for use as active packaging." ..... Author should discuss about what is the use of provitamin A β-carotene in the developed sheet?? ... Why it is important for food packaging??. ... Author can discuss these aspects in the introduction section. 

- the effects of addition tucumã oil 20 (TO) and TO microparticles on antimicrobial properties of the film should be discussed. 

Author Response

Reviewer 1: The manuscript entitled "Characterization of cassava starch extruded sheets incorporated with tucumã oil microparticles" can be improved by addressing the suggestions given below :-

1) Line no.- 24, "The sheets enriched with TO (FO) and the micro- 24 particles (FSD, FSC, and FSDC)" ....  the abbreviations are not clear

Answer: The abbreviations were clarified, as follows:

Lines 24-25: “The sheets enriched with tucumã oil (FO) and the microparticles produced by SD, SC and SDC (FSD, FSC, and FSDC, respectively)…”

2) Line no.- "One may conclude that promising active packages were obtained by extrusion of cassava 30 starch sheets added with pure and encapsulated TO" ... how authors can say the developed sheet is active??? ...Which active properties they have tested?? ...

The release of oil from the sheet/oil microparticles should be included and/or any active function of the film

Answer: Thank you for the comment. In fact, any active properties nor the release of oil from the sheets have been tested in the present study, as indicated in the Conclusions. The focus of this work was to assess the feasibility of introducing different types of microparticles into extruded sheets by characterizing their mechanical, optical and physical properties, since few works investigate the production of enriched films/sheets by extrusion. Knowing that such incorporation results in promising materials, further studies are needed to evaluate the active properties displayed by extruded cassava starch sheets added with tucumã oil microparticles. To avoid any misunderstanding, the term “promising” was replaced by “potential” in the Abstract. In addition, the following information was included in the Introduction:

Lines 64-76: “Unfortunately, both the fruit and oil have been poorly explored at the research and indus-trial level, and the capacity of such materials to contribute for food preservation was not investigated so far. However, β-carotene is known to present antioxidant activity in food systems and has been applied for production of active films, resulting in antioxidant packages, which improved the oxidative stability of sunflower oil and fried peanuts dur-ing storage [6][19][20]. In addition, β-carotene is a vitamin A precursor [22] and its appli-cation into edible coatings could improve the nutritional value of the packaged product, as suggested by recent works in which edible films enriched with vitamin D3 were obtained [23][24]. Considering that β-carotene is the major carotenoid in TO [25] and that this compound presents provitamin A activity and strong antioxidant capacity, the addition of TO to films and sheets has potential to provide similar active and functional properties to food packages.”

3) Figure 2, should be improved by placing 2 or 3 pictures in a row

Answer: The correction was implemented.

4) Line no.- 488 "Nevertheless, the addition of an oil rich in provitamin A β-carotene, in free and microencapsulated form, to extruded sheets based on biopolymers is reported here for the first time, and the obtained materials have promising attributes for use as active packaging." ..... Author should discuss about what is the use of provitamin A β-carotene in the developed sheet?? ... Why it is important for food packaging??. ... Author can discuss these aspects in the introduction section.

Answer: Thank you for the comment. The following was added to the Introduction to address it:

Lines 71-76: “In addition, β-carotene is a vitamin A precursor [22] and its application into edible coat-ings could improve the nutritional value of the packaged product, as suggested by recent works in which edible films enriched with vitamin D3 were obtained [23][24]. Considering that β-carotene is the major carotenoid in TO [25] and that this compound presents provitamin A activity and strong antioxidant capacity, the addition of TO to films and sheets has potential to provide similar active and functional properties to food packages.”

5) the effects of addition tucumã oil 20 (TO) and TO microparticles on antimicrobial properties of the film should be discussed.

Answer: As mentioned in a previous question, the antimicrobial activity of the sheets enriched with tucumã oil and its microparticles was not determined in this work. Actually, tucumã oil is not supposed to possess such activity, since its major bioactive compound (beta-carotene) does not present antimicrobial capacity, but strong antioxidant property instead. This was indicated in a paragraph added to the Introduction, as follows:

Lines 64-76: “Unfortunately, both the fruit and oil have been poorly explored at the research and indus-trial level, and the capacity of such materials to contribute for food preservation was not investigated so far. However, β-carotene is known to present antioxidant activity in food systems and has been applied for production of active films, resulting in antioxidant packages, which improved the oxidative stability of sunflower oil and fried peanuts dur-ing storage [6][19][20]. In addition, β-carotene is a vitamin A precursor [22] and its appli-cation into edible coatings could improve the nutritional value of the packaged product, as suggested by recent works in which edible films enriched with vitamin D3 were obtained [23][24]. Considering that β-carotene is the major carotenoid in TO [25] and that this compound presents provitamin A activity and strong antioxidant capacity, the addition of TO to films and sheets has potential to provide similar active and functional properties to food packages.”

Reviewer 2 Report

The presented work is well structured, clear and its relevance is supported by infiormation in the introduction. 

Methodology is clearly explained and adequate to objectives and findings. I would only recommend:

- a clarification between the concepts "films" and "sheets" 

- references to, and ideally data regarding how encapsulation impacts oil functionality in food conservation as this greatly impacts conclusions of this study regarding technology application in active food packaging

-avoiding subjective terms (e.g. line 368: "a lit bit lower", line 470 "curious", line 477 "attractive"

Author Response

Reviewer 2: The presented work is well structured, clear and its relevance is supported by infiormation in the introduction. Methodology is clearly explained and adequate to objectives and findings. I would only recommend:

1) a clarification between the concepts "films" and "sheets"

Answer: The following sentence was added to the Introduction to address this suggestion:

Lines 38-39: “Depending on the thickness of the package layer, packaging materials can be classified as films (thickness < 0.25 mm) or sheets (thickness > 0.25 mm) [2].”

2) references to, and ideally data regarding how encapsulation impacts oil functionality in food conservation as this greatly impacts conclusions of this study regarding technology application in active food packaging

Answer: The following paragraph was included in the Results and Discussion to address the comment:

Lines 306-320: “The improved stability of sensitive bioactives like carotenoids after encapsulation is crucial for later application of the obtained particles. As mentioned above, carotenoids are mainly degraded by oxidation, which is triggered by high temperatures, oxygen, light, among other factors [39]. In addition, β-carotene is the major bioactive compound in tu-cumã oil that could provide active properties to sheets for food packaging purpose, as discussed before. Previous studies demonstrated the impact of encapsulation on preserv-ing the properties of materials with relevance in terms of food preservation. For example, spray drying improved the oxidative stability of tucumã oil during exposition to high temperatures (similar to the ones applied during extrusion) by at least 142-fold, suggest-ing that encapsulation maintained the oil/carotenoids properties of interest for food preservation [17]. In another work, cassava starch films containing nanoencapsulated β-carotene were more efficient in preventing sunflower oil oxidation during storage (per-oxide value of 6.72 mEq/kg) compared to non-encapsulated β-carotene (peroxide value of 9.31 mEq/kg), due to the improved stability of the encapsulated carotenoid and the higher light-barrier property of films added with the particles [6].”

3) avoiding subjective terms (e.g. line 368: "a lit bit lower", line 470 "curious", line 477 "attractive"

Answer: The corrections were implemented.

Reviewer 3 Report

The articles is of high quality and importance.

Line 50: consider changing "may" with "make"

Line 244: Even 5 kV is a low voltage, did the authors observe some changes in the sample (melting of the oil pariticles)?

Line 298: Is the word "aspects" the most appropriate one?

Author Response

Reviewer 3: The articles is of high quality and importance.

1) Line 50: consider changing "may" with "make"

Answer: The correction was implemented.

2) Line 244: Even 5 kV is a low voltage, did the authors observe some changes in the sample (melting of the oil pariticles)?

Answer: In case the reviewer is referring to changes taking place during microscopy analysis, unfortunately the authors cannot answer that, since the analysis was performed by a skilled technician. However, the authors believe that the voltage applied during microscopy could not cause any changes to the particles incorporated into the sheets, since not even the harsh extrusion conditions could melt the lipidic particles, as discussed in the manuscript. In addition, this voltage was applied in many previous studies of our group and no changes were observed during the microcopy analysis.

3) Line 298: Is the word "aspects" the most appropriate one?

Answer: The word “aspects” was replaced by “appearances”.

Round 2

Reviewer 1 Report

In the manuscript entitled "Characterization of cassava starch extruded sheets incorporated with tucumã oil microparticles" author have done significant changes in the manuscript, as per my suggestions.